# Lung Ultrasound B-lines Occurrence in Relation to Left Ventricular Function and Hydration Status in Hemodialysis Patients

**DOI:** 10.3390/medicina55020045

**Published:** 2019-02-12

**Authors:** Agnieszka Pardała, Mariusz Lupa, Jerzy Chudek, Aureliusz Kolonko

**Affiliations:** 1Dialysis Centre, Fresenius Nephrocare, 34-600 Limanowa, Poland; agnieszka.pardala@fmc-ag.com; 2Department of Internal Medicine, District Hospital, 34-600 Limanowa, Poland; mariuszlupa@poczta.onet.pl; 3Department of Internal Medicine and Oncological Chemotherapy, Medical University of Silesia, 40-027 Katowice, Poland; chj@poczta.fm; 4Department of Nephrology, Transplantation and Internal Medicine, Medical University of Silesia, 40-027 Katowice, Poland

**Keywords:** bioimpedance analysis, echocardiography, chronic kidney disease, left ventricular ejection fraction, lung comets, lung ultrasound, overhydration

## Abstract

Background and objective: Reliable assessment of the fluid status in hemodialysis (HD) patients is often difficult. A lung ultrasound with an assessment of the B-lines (“lung comets” (LCs)) number is a novel hydration status measure. However, the occurrence of left ventricular dysfunction may have a significant effect on pulmonary congestion and further modulate the LC number. The aim of this study was to analyze to what extent left ventricular dysfunction, pulmonary hypertension, and hypervolemia affect the occurrence of LC in a cohort of prevalent HD patients. Material and methods: This cross-sectional study included 108 assessments performed in 54 patients who attended thrice weekly outpatient HD. Each patient’s fluid status was evaluated twice, prior to HD sessions, using echocardiography, LC number assessment, measurement of inferior vena cava (IVC) diameters, and bioelectric impedance analysis (BIA). Patients were stratified into three subgroups according to their LC number. Results: There were 76 separate assessments with mild (<14), 16 with moderate (14–30), and 16 with severe (>30) LC occurrence. There was a negative correlation between the LC number and left ventricular ejection fraction (LVEF), and positive correlations between the LC number and mitral gradient, and the left and right atrium area and volume, but not with the BIA-derived relative fluid overload. Multivariate linear regression analysis revealed that the LC number was proportionally related to the mitral gradient (β = 0.407 (0.247–0.567), p < 0.001) and IVC max diameter (β = 0.219 (0.060–0.378), p < 0.01), and was inversely related to LVEF (β = −0.431 (−0.580 to −0.282), p < 0.001). Conclusions: The number of LCs appears to reflect both overhydration and left ventricular dysfunction in our HD patients cohort. Therefore, heart failure must be considered as an important factor limiting the usefulness of LCs number assessment in this population.

## 1. Introduction

Chronic volume overload, either clinically apparent or subclinical, is common in hemodialysis (HD) patients and often leads to the development of resistant hypertension, cardiomyopathy, and heart failure [1]. These complications contribute to the several times greater cardiovascular mortality among HD patients compared with the general population [2,3]. In the majority of maintenance HD patients, precise assessment and effective management of fluid overload remains a serious challenge. As the patient’s dry weight determined using clinical parameters is often inadequate, different traditional and novel fluid status measures have been proposed, including chest X-ray, ultrasound measurement of inferior vena cava (IVC) diameters, echocardiography, bioelectric impedance analysis (BIA), and the assessment of natriuretic peptide levels [4,5].

Recently, the detection of extravascular lung water by chest ultrasound was proposed [6] and subsequently validated [7,8] as a reliable noninvasive tool for quantitative measurement of fluid overload and pulmonary congestion in HD patients. The number of B-lines (“lung comets” (LCs)) counted during the lung ultrasound positively correlated with interdialytic weight gain, New York Heart Association (NYHA) classification, and pulmonary artery systolic pressure in a multivariate linear regression analysis [9]. Nevertheless, there are conflicting reports on the effects of arteriovenous fistula (AVF) creation and flow on the development of pulmonary hypertension within a long-term period [10,11]. Beigi et al. [11] found a negative correlation between pulmonary artery pressure and left ventricular ejection fraction (LVEF). On the other hand, AVF creation increased LVEF [12]. However, there is limited evidence concerning the relationship between LCs score and LVEF with reference to the hydration status in HD patients. Zoccali et al. [7,13] reported a relationship between the number of LCs and LVEF both in HD and peritoneal dialysis patients, but the estimation of the fluid status was based on BIA only. Such an association has also been studied in patients with acute and chronic heart failure [14,15], as well as in patients with acute pulmonary edema [16], yielding inconclusive results. 

Thus, the aim of this study was to analyze to what extent left ventricular dysfunction, pulmonary hypertension, and hypervolemia affect the occurrence of ultrasound-derived LCs number in a cohort of prevalent HD patients. 

## 2. Methods

### 2.1. Study Population

This cross-sectional study included 108 assessments performed in 54 patients with chronic kidney disease (CKD) who attended thrice weekly HD in one dialysis center for at least 3 months and agreed to participate. The study was performed between July 2015 and October 2017. Fresenius 4008 machines and high-flux dialyzers (Fx Cordiax, Fresenius Medical Care, Bad Homburg, Germany) were used for all patients. Patients with active infection, hepatic cirrhosis, cancer, or medical contraindication for bioelectrical impedance measurements, as well as patients with significant valve defects (including moderate to severe mitral or tricuspid valve insufficiency or stenosis) were excluded from the study.

The study protocol was accepted by the Bioethics Committee of the Medical University of Silesia in Katowice (KNW/0022/KB1/47/14), and all participants provided written informed consent. The study was conducted in accordance with the Declaration of Helsinki. In addition to data retrieved from the center-operated registry, we performed a carotid ultrasound with intima-media thickness (IMT) measurements in each patient. Then, we evaluated the patients’ fluid status using echocardiography, chest ultrasound with assessment of the B-lines number, and the IVC diameter twice: Immediately prior to the first and third dialysis session in the same week. At the same time-points, BIA was performed and blood samples were withdrawn. All echocardiography examinations were performed by one cardiologist (ML), and the rest of the above ultrasound-based measurements were performed by second investigator (AP).

### 2.2. Clinical and Anthropometric Measurements

Body weight and height were measured following standard procedures, and body mass index (BMI) was calculated (kg/m^2^). Body surface area (BSA) was calculated according to the DuBois formula (0.20247 × weight (kg)^0.425^ × height (m)^0.725^) and expressed in m^2^.

Blood pressure was measured during the patients’ examination immediately before each dialysis session. A subgroup of patients with a high risk of pulmonary hypertension was defined as having calculated right ventricular systolic pressure (RVSP) >35 mmHg.

### 2.3. Laboratory Measurements

Plasma N-terminal prohormone for brain natriuretic peptide (NT-proBNP) concentration was measured by the electrochemiluminescence method using a commercially available Cobas E411 analyzer (Roche Diagnostics GmbH, Mannheim, Germany) with intermediate precision <4.6%.

### 2.4. Carotid Artery IMT

Carotid ultrasound was performed using a Toshiba Xario machine (Toshiba Medical Systems Corporation, Tochigi, Japan) equipped with a 7.5 MHz linear transducer. The evaluation included the common, internal, and external carotid arteries and the carotid bifurcation on each side. The common carotid artery IMT was measured proximal to the carotid bulb, approximately every 1 cm, omitting visible plaques. The results from three separate measurements on each side were then averaged.

### 2.5. Echocardiography

Echocardiographic measurements were performed using Hitachi Aloka ProSound Alpha 6, equipped with 1—15 MHz cardio transducer UST 5299 (Hitachi Aloka Medical, Ltd., Tokyo, Japan). Full M-mode and two-dimensional measurements were performed as recommended by the American Society of Echocardiography [17]. These measurements included left ventricular end-diastolic and end-systolic diameters, intraventricular septum, and posterior wall end-diastolic thickness. LVEF was calculated by using the biplane Simpson method. Left (LA) and right atrium (RA) volumes were calculated according to the Simpson method. RVSP was calculated from the apical 4-chamber view as a measured velocity of the tricuspid backflow wave with an added 10 mmHg for the approximate right ventricle pressure. Left ventricular mass (LVM) was calculated according to the Devereux formula [18]. LVM was indexed for BSA.

### 2.6. Lung Ultrasound

Lung ultrasound examinations were performed using a Toshiba Xario machine equipped with a 3.5 MHz convex transducer. Patients were in the supine position. The probe was placed along the intercostal spaces (II-V on the right side and II-IV on the left side) in mid-axillary, anterior axillary, mid-clavicular, and parasternal lines. The number of B-lines was assessed for all locations and the sum was calculated for a given patient denoting the extent of extravascular fluid in the lungs. At each scanning site, B-lines were counted from zero to ten, with zero indicating a complete absence of B-lines, and a full white screen corresponding to ten B-lines [19].

### 2.7. IVC Diameter Measurement

A Toshiba Xario machine equipped with a 3.5 MHz convex transducer was used in all examinations. IVC diameters were measured in the supine position, using a sagittal section of the IVC. The IVC diameter was measured at end-expiration (maximal diameter) and at end-inspiration (minimal diameter) at 1 cm caudal of the hepatic vein.

### 2.8. Bioelectrical Impedance (BIA)

The hydration status was estimated with a Body Composition Monitor (BCM), produced by Fresenius Medical Care^®^ (Fresenius Medical Care AG & Co. D-61366 Bad Homburg, Germany). The multifrequency BCM is a non-invasive method to estimate the intracellular and extracellular water volume using bioimpedance spectroscopy at 50 frequencies (5 kHz to 1000 MHz). The relative fluid overload (RFO), i.e., hydration normalized to extracellular water, allows the comparison of hydration status regardless of the patients’ weight, height, gender, or age. The normal hydration status is defined by RFO from −7% to 7%, corresponding to the 10th and 90th percentiles of a healthy population [20]. In our cohort, BIA assessments were performed immediately prior to the dialysis session.

### 2.9. Statistical Analysis

Statistical analyses were performed using the STATISTICA 12.0 PL for Windows software package (StatSoft Polska, Kraków, Poland) and MedCalc 12.3.0.0. (MedCalc Software, Mariakerke, Belgium). The values were presented as mean values and 95% confidence intervals (CIs), median values with first and third quartiles, or frequencies for qualitative data. The main comparison was performed in three subgroups, defined based on the total number of B-lines: <14, 14–30, and >30. Differences in the distribution of qualitative variables between three study subgroups were compared by χ2 and χ2 for a trend, whereas that of quantitative variables were compared by an analysis of variances (*t*-test ANOVA) or ranks (Mann-Whitney U or Kruskal-Wallis test). Correlation coefficients were calculated using the Pearson test (due to nonparametric distribution, the number of LCs and values of plasma NT-pro-BNP concentration were previously logarithmically transformed). To compare the results of different fluid status measures obtained in two groups, Student’s *t*-test was used. Multivariate linear forward stepwise regression analysis was performed for the number of LCs as a dependent variable and the potential explanatory variables, selected based on univariate analyses (IVC max, LVEF, RVSP, mitral gradient, LA and RA area). In all statistical tests, ‘*p*’ values below 0.05 were considered statistically significant.

## 3. Results

### 3.1. Study Group

The study group comprised 22 female and 32 male patients with the following characteristics: Mean age, 58.2 (95% CI: 53.7–62.3) years; mean BMI, 25.9 (24.4–27.3); and mean dialysis vintage, 47 (32–63) months. The causes of CKD were as follows: Glomerular disease (25.9%), hypertensive or ischemic nephropathy (25.8%), pyelonephritis (20.4%), polycystic kidney disease (13%), diabetic nephropathy (9.3%), and other or unknown (5.6%). The comorbidities included: Hypertension (92.6%), diabetes mellitus (27.8%), and ischemic heart disease (57.4%). Major adverse cardiovascular events, including stroke, myocardial infarction, and coronary artery stenting or bypass graft, were previously diagnosed in 33.3% of study participants. In total, 44.4%, 27.8%, 18.5%, and 9.3% of patients were in NYHA classes I, II, III, and IV, respectively. At the time of the study, 11.1% of participants were active smokers.

### 3.2. Comparison of LC Score Categories

Among all 108 separate assessments performed in 54 patients, there were 76 measurements classified as mild (<14), 16 as moderate (14–30), and 16 as severe (>30) LC scores, based on the total number of B-lines in the lung sonography (7). The mean B-lines intra-patient variability was 8 (95% CI 5–10). The above LC categories did not differ significantly with respect to age, dialysis vintage, residual diuresis, or type of vascular access (arterio-venous fistula vs. catheter). The analysis of sonographic data revealed a similar left ventricular mass index (LVMI), but decreasing LVEF and increasing LA area and volume as well as mitral valve gradient across the LC score categories (Table 1). Less pronounced differences were observed for the RA area and volume. Individuals with moderate to severe LC scores were characterized by a high occurrence (>50%) of RVSP >35 mmHg and increased plasma concentration of NT-proBNP (Table 1). Assessments with LC >30 exhibited increased minimal and maximal VCI diameters in comparison to mild and moderate LC score categories. The difference was greater for minimal than maximal IVC diameters (40.2% vs. 29.0%). There was no difference in RFO measured by BIA in the analyzed LC categories.

The log LCs number was inversely related to LVEF (r = −0.443, *p* < 0.001). We also found positive correlations between the log LCs number and LA area and volume (r = 0.392, *p* < 0.001 and r = 0.380, *p* < 0.001, respectively) as well as the mitral gradient (r = 0.326, *p* = 0.001). Similar correlations with the RA area and volume were less pronounced (r = 0.248, *p* = 0.012 and r = 0.237, *p* = 0.018, respectively). There was also a significant positive correlation between the log LCs number and IVC max diameter (r = 0.221, *p* = 0.025), and a borderline association with the IVC min diameter (r = 0.176, *p* = 0.075). Correlations between the log LCs number and IVC diameters indexed for BSA yielded similar results. Importantly, we found a positive correlation between the log LCs number and RVSP (r = 0.222, *p* = 0.024). Accordingly, the percentage of patients that fulfilled the sonographic criteria of pulmonary hypertension presented a significant trend across the LC score categories (*p* = 0.02 for the trend). Of note, we also found a significant correlation between the log LCs number and log NT-proBNP (r = 0.455, *p* < 0.001). Finally, there was no association between the LCs number and RFO.

### 3.3. A Comparison of the Assessments Scored According to LVEF ≥ or < 50%, RVSP Value >35 mmHg, and BIA-Derived RFO > or ≤ 7%

The core characteristics of the analyzed subgroups and their results of the fluid status measurements are given in Table 2. The number of LCs was significantly higher in assessments with EF < 50% (Figure 1).

Additionally, correlation analyses revealed that RVSP was significantly associated with LA and RA areas and volumes (LA: r = 0.311 and r = 0.334, both *p* < 0.001; RA: r = 0.193, *p* = 0.049 and r = 0.351, *p* < 0.001, respectively), as well as with LA and RA volumes indexed for BSA (r = 0.385 and r = 0.396, both *p* < 0.001, respectively). Additionally, RVSP was also correlated with the mitral gradient (r = 0.416, *p* < 0.001), log NT-proBNP (r = 0.211, *p* = 0.034), and both IVC diameters (IVC min: r = 0.237, *p* = 0.016; IVC max: r = −0.365, *p* < 0.001), which were also apparent after indexing for BSA (r = 0.288, *p* = 0.003 and r = 0.430, *p* < 0.001, respectively). Of note, both LVEF (*p* = 0.38) and RFO (*p* = 0.30) did not correlate with RVSP. Importantly, there was no association between the RFO and IVC diameters, LC number, RVSP, mitral gradient, LA and RA areas and volumes, LVEF, and LVMI (data not shown). Of note, RFO was significantly correlated with log NT-proBNP (r = 0.254, *p* = 0.01).

### 3.4. Multivariate Linear Regression Analysis of the LC Number

The multivariate forward stepwise regression model revealed that the LCs number was proportional to the mitral gradient (β = 0.407 (0.247–0.567), *p* < 0.001) and IVC max diameter (β = 0.219 (0.060–0.378), *p* < 0.01), and inversely related to LVEF (β= −0.431 (−0.580 to −0.282), *p* < 0.001). This model explained 40% of all the LCs number variability.

## 4. Discussion

As early as 2010, Mallamaci et al. [7] published a landmark paper, where they showed that in HD patients, lung water assessed by chest ultrasound was strongly associated with several measures of cardiac performance, including LVEF. However, patients’ fluid status was defined based solely on pre-dialysis BIA and the NYHA class. Moreover, this study did not analyze the potential influence of residual diuresis or the type of vascular access on the LCs number. Finally, there was no information concerning the presence of clinically important valve insufficiency in the analyzed cohort [7].

Chronic expansion of the extracellular volume, together with its consequences (poorly controlled hypertension, left ventricular hypertrophy, and heart failure), is a universal component of end stage kidney disease [21]. It was recently shown that chronic pre-dialysis fluid overload predicted excessive risk of mortality across all blood pressure categories, including SBP < 130 mmHg [22,23]. However, in the majority of patients, fluid overload is asymptomatic, and its diagnosis remains difficult [24]. Moreover, unlike antihypertensive medications, a strict volume control strategy provides optimal blood pressure control, however, its application remains limited because of several factors, including the lack of a gold standard method to assess the volume status [25]. A substantial disagreement between different techniques could be explained by the fact that changes in fluid status may not occur in different body compartments in parallel, e.g., by the influence of hypoalbuminemia, an increase in vascular permeability, or the co-occurrence of heart failure [24]. Of note, a week association of lung water measured by sonography and total body water assessed by BIA further indicates that lung congestion can only partly be explained by volume overload [7,21,26]. It seems that left ventricular disorders play a major role in lung congestion in HD patients [21].

In this study, we showed a significant negative relationship between the number of pulmonary B-lines detected by lung ultrasound and LVEF in maintenance HD patients, after including all known potential covariates of the LCs number and cardiac function. We also found a greater rate of higher LC categories in participants with LVEF < 50%. Moreover, the reliability of the LC score in the assessment of the degree of pulmonary congestion was proven by the association of the LC number with the LA area and volume, mitral gradient, RVSP, and IVC max diameter, whereas the independent relation of the LCs number and LVEF was confirmed by multivariate linear regression analysis. By contrast, we did not endorse the utility of BIA in the fluid status assessment in our cohort. The results of other studies concerning the efficacy of BIA in HD patients are inconsistent [27,28,29,30]. Of note, a recent study showed that BIA performs better than the LCs number as an additional predictor in a Cox survival analysis model, however, the patient age was not included in this analysis, despite an almost 5-year between-group difference [30]. Moreover, BIA results’ dependence on recent physical activity, eating, or fluid intake prior to examination limited its usefulness in HD patients [5]. Thus, combining BIA with the NT-proBNP concentration or other biomarkers may be more helpful for dry weight assessment [31]. 

The strength of our study is that the fluid status assessment in HD patients was performed using all accepted ultrasound-based methods, including the LC number, concomitantly with BIA analysis. Moreover, all examinations were performed by one researcher, except the echocardiographic studies. Also, patients with diagnosed valve defects, including moderate to severe mitral or tricuspid valve insufficiency or stenosis, were excluded. Lastly, we also recorded and analyzed the residual diuresis and the type of vascular dialysis access as potential covariates of the LCs number, fluid status, and cardiac performance. 

A study limitation is the small number of measurements in patients with markedly reduced LVEF (16.7% with LVEF < 50%, including 6.5% with LVEF < 40%).

## 5. Conclusions

In summary, the number of LCs reflected both overhydration and left ventricular dysfunction in our HD patients cohort. Therefore, heart failure should be considered as an important factor limiting the usefulness of the LCs number assessment in this population.

## Figures and Tables

**Figure 1 medicina-55-00045-f001:**
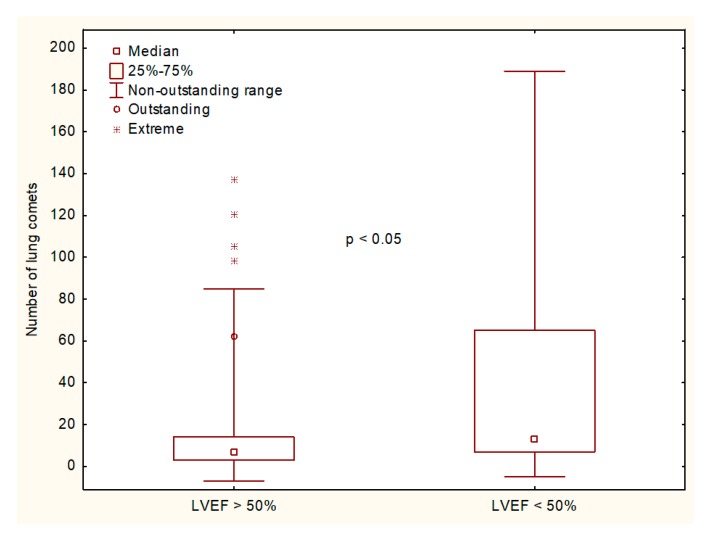
The comparison of the lung comets (LC) number in hemodialysis patients with left ventricular ejection.

**Table 1 medicina-55-00045-t001:** The demographic, anthropometric, and clinical characteristics of three study subgroups, with imaging measurements in these subgroups, based on the number of B-lines scored during the lung ultrasound.

	Lung Comets Score			Statistics			
	<14 Group 1 *n* = 76	14–30 Group 2 *n* = 16	>30 Group 3 *n* = 16	ANOVA/Chi^2^	1 vs. 2	1 vs. 3	2 vs. 3
Age (years)	58 (54–62)	56 (45–68)	62 (56–68)	0.60	-	-	-
Gender (M/F)	48/28	9/7	7/9	0.15	-	-	-
Dialysis vintage (mo) *	29 (7–61)	61 (5–96)	21 (8–101)	0.65 **	-	-	-
Vascular access AVF (*n*, %) Catheter (*n*, %)	48 (63.2) 28 (36.8)	8 (50.0) 8 (50.0)	10 (62.5) 6 (37.5)	0.61	-	-	-
Residual diuresis (mL) *	500 (250–1000)	450 (50–850)	500 (50–1000)	0.53 **	-	-	-
BMI (kg/m^2^)	25.5 (24.4–26.6)	24.9 (21.3–28.5)	28.7 (25.6–31.8)	0.07	-	-	-
IMT (mm)	0.69 (0.67–0.72)	0.66 (0.61-0.71)	0.70 (0.64–0.76)	0.57	-	-	-
LVMI	165 (152–177)	174 (137–210)	169 (143–195)	0.83	-	-	-
LVEF (%)	58.4 (56.6–60.3)	53.1 (47.2–59.0)	47.8 (40.4–55.2)	<0.001	0.26	0.006	0.26
RVSP (mm Hg)	32.7 (30.9–34.5)	36.1 (31.5–40.8)	38.1 (33.4–42.9)	0.03	0.45	0.15	0.77
RVSP > 35 mmHg (*n*, %)	27 (35.5)	9 (56.2)	10 (62.5)	0.02	0.12	0.047	0.72
LAVI	30.2 (27.9–32.4)	33.3 (26.3–40.4)	40.1 (31.9–48.3)	0.007	0.72	0.04	0.23
RAVI	18.6 (17.1–20.1)	21.2 (17.1–26.4)	24.6 (15.6–33.4)	0.04	0.61	0.15	0.66
Mitral valve gradient (mmHg)	3.3 (3.0–3.6)	4.3 (2.5–6.1)	5.4 (2.4–8.5)	0.02	0.58	0.08	0.48
IVC min/BSA	4.7 (4.3–5.1)	4.5 (3.5–5.5)	6.3 (5.1–7.5)	0.005	0.96	0.04	0.02
IVC max/BSA	7.1 (6.7–7.5)	7.3 (6.1–8.5)	8.9 (7.9–9.9)	0.002	0.93	0.02	0.04
RFO	7.9 (6.2–9.5)	9.3 (5.5–13.1)	9.8 (7.2–12.4)	0.52	-	-	-
NT-proBNP (pg/mL) *	2695 (1551–5231)	8173 (4259–14264)	8559 (3832–44582)	<0.001 **	0.08	<0.001	0.55

Data presented as means and 95% CI or frequencies, except * median value with first and third quartiles. ** ANOVA after data logarithmization. AVF, arterio-venous fistula; BMI, body mass index; BSA, body surface area; IMT, intima-media thickness; LVMI, left ventricular mass indexed for BSA; EF, left ventricular ejection fraction; RVSP, right ventricular systolic pressure; LAVI, left atrial volume indexed for BSA; RAVI, right atrial volume indexed for BSA; IVC, inferior vena cava; RFO, relative fluid overload calculated by a Body Composition Monitor; NT-proBNP, N-terminal prohormone for brain natriuretic peptide.

**Table 2 medicina-55-00045-t002:** Demographics and the results of the diagnostic imaging measurements performed prior to the hemodialysis session in patients with left ventricular ejection fraction (LVEF) ≥50% and <50% (left panel), in patients with right ventricle systolic pressure ≤35 and >35 mmHg (middle panel), and in patients with relative fluid overload (RFO) >7% or ≤7%, as measured prior to the dialysis session using electrical bioimpedance (right panel).

Title	LVEF ≥ 50% *n* = 90	LVEF < 50% *n* = 18	*p*	RVSP ≤ 35 mmHg *n* = 62	RVSP > 35 mmHg *n* = 46	*p*	RFO ≤ 7% *n* = 38	RFO > 7% *n* = 70	*p*
Age (years)	57.4 (53.7–61.0)	62.7 (58.2–67.2)	0.21	58.1 (54.1–62.1)	58.4 (53.1–63.7)	0.94	59.3 (54.9–63.7)	57.7 (53.3–62.1)	0.63
Gender (M/F)	54/36	10/8	0.73	40/22	24/22	0.20	21/17	44/26	0.44
BMI (kg/m^2^)	25.6 (24.5–26.7)	27.4 (24.7–30.0)	0.20	26.0 (24.7–27.3)	25.7 (24.0–27.4)	0.77	25.9 (24.4–27.5)	25.7 (24.3–27.0	0.80
Dialysis vintage (mo) *	22 (7–68)	59 (12–96)	0.26 **	47 (7–77)	21 (7–52)	0.22 **	34 (7–61)	28 (6–74)	0.59 **
Vascular access AVF (*n*, %) Catheter (*n*, %)	56 (62.2) 34 (37.8)	10 (55.6) 8 (44.4)]	0.60	39 (62.9) 23 (37.1)	27 (58.7) 19 (41.3)	0.66	25 (65.8) 13 (34.2)	42 (60.0) 28 (40.0)	0.56
EDD (mm)	48.6 (47.5–49.7)	57.0 (55.1–58.9)	<0.001	50.0 (48.4–51.6)	50.0 (48.4–51.5)	0.93	49.5 (47.7–51.4)	50.1 (48.7–51.6)	0.61
ESD (mm)	29.4 (28.5–30.3)	40.1 (37.8–42.3)	<0.001	31.4 (29.8–33.0)	30.9 (29.3–32.5)	0.69	31.4 (29.6–33.2)	30.9 (29.5–32.3)	0.65
IVS (mm)	13.3 (12.7–13.8)	13.4 (12.6–14.3)	0.80	13.3 (12.6–14.0)	13.3 (12.7–14.0)	1.00	12.8 (12.1–13.4)	13.6 (12.9–14.2)	0.12
PW (mm)	12.0 (11.4–12.6)	12.1 (11.2–13.0)	0.85	11.8 (11.1–12.6)	12.2 (11.5–12.9)	0.48	11.5 (10.7–12.4)	12.3 (11.6–13.0)	0.17
LVM (g)	295 (271–319)	384 (340–429)	0.003	309 (277–340)	311 (281–342)	0.90	287 (254–319)	322 (292–352)	0.13
LVMI	158 (147–170)	207 (178–236)	<0.001	164 (148–179)	170 (156–185)	0.56	154 (140–168)	173 (158–188)	0.09
LVH (%)	78.9	94.4	0.12	79.0	84.8	0.45	86.8	78.3	0.28
LVEF (%)	60.0 (58.5–60.8)	38.1 (33.1–43.0)	<0.001	55.1 (52.4–57.7)	57.4 (54.5–60.3)	0.25	56.7 (54.4–59.0)	56.2 (53.5–58.8)	0.78
RVSP	34.3 (32.5–36.0)	32.7 (28.6–36.8)	0.47	28.4 (27.1–29.7)	41.5 (40.0–43.1)	<0.001	33.4 (31.1–35.6)	34.5 (32.4–36.7)	0.48
LAVI	30.7 (28.4–33.0)	39.0 (32.7–45.4)	0.005	29.5 (27.0–32.1)	35.6 (31.7–39.5)	0.008	31.0 (27.3–34.8)	32.6 (29.7–35.6)	0.51
RAVI	19.1 (17.6–20.5)	24.2 (16.1–32.3)	0.03	16.8 (15.4–18.3)	24.1 (20.7–27.5)	<0.001	19.2 (16.9–21.6)	20.5 (17.9–23.0)	0.52
Mitral valve gradient (mmHg)	3.8 (3.2-4.5)	3.5 (2.5–4.5)	0.64	3.1 (2.8–3.5)	4.7 (3.5–5.8)	<0.001	3.5 (2.6–4.5)	3.9 (3.2–4.6)	0.48
Lung comets LC 0–13 (*n*, %) LC 14–30 (*n*, %) LC > 30 (*n*, %)	67 (74.4) 13 (14.4) 10 (11.1)	9 (50.0) 3 (16.7) 6 (33.3)	0.04	49 (79.0) 7 (11.3) 6 (9.7)	27 (58.7) 9 (19.6) 10 (21.7)	0.07	28 (73.7) 6 (15.8) 4 (10.5)	49 (70.0) 10 (14.3) 11 (15.7)	0.76
IVC min/BSA	4.8 (4.4–5.1)	5.6 (4.2–7.0)	0.1	4.4 (3.9–4.9)	5.6 (5.1–6.1)	0.001	4.7 (4.1–5.3)	5.0 (4.5–5.4)	0.48
IVC max/BSA	7.4 (7.0–7.8)	7.6 (6.3–8.8)	0.73	6.7 (6.3–7.2)	8.3 (7.8–8.9)	<0.001	7.3 (6.7-7.9)	7.4 (7.0–7.9)	0.67
RFO	8.3 (6.8–9.8)	8.7 (5.2–12.2)	0.84	8.0 (6.2–9.8)	8.8 (6.8–10.9)	0.52	1.3 (−0.4–3.0)	12.2 (11.2–13.2)	<0.001
NT-proBNP (pg/mL) *	2856 (1601–5667)	13316 (5579–66120)	<0.001 ***	2810 (1448–9311)	4556 (2223–9324)	0.17 ***	3203 (1331–6255)	3794 (1880–12024)	0.07 ***

Data presented as means and 95% CI, except * median value and interquartile range. ** ANOVA after data logarithmization. *** *t*-student test after data logarithmization. BMI, body mass index; AVF, arterio-venous fistula; EDD, end-diastolic diameter; ESD, end-systolic diameter; IVS, intraventricular septum; PW, posterior wall; LVM, left ventricular mass; LVMI, left ventricular mass indexed for BSA; LVH, left ventricular hypertrophy; LVEF, left ventricular ejection fraction; RVSP, right ventricular systolic pressure; LAVI, left atrial volume indexed for BSA; RAVI, right atrial volume indexed for BSA; LC, lung comets; IVC, inferior vena cava; RFO, relative fluid overload calculated by a Body Composition Monitor prior to the hemodialysis session; NT-proBNP, *N*-terminal prohormone for brain natriuretic peptide.

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
