# Peer review of "Lung Ultrasound B-lines Occurrence in Relation to Left Ventricular Function and Hydration Status in Hemodialysis Patients"

_medicina, 2019, doi:10.3390/medicina55020045_

Round 1
Reviewer 1 Report
The manuscript “Lung ultrasound B-lines occurrence in relation to left ventricular function and hydration status in hemodialysis patients” presents the results of the study carried out with the aim to evaluate to what extent left ventricular dysfunction, pulmonary hypertension and hypervolemia affect the occurrence of ultrasound-derived LCs in hemodialysis patients.
I have a few minor remarks.
1. It has not been explained on the basis of which three categories of LC scores are defined (median, IQR or?).
2. In table 2 it would be better if the title of the column called “Statistics” be “p”.
3. Would not it be better to present the results of the correlation described in the text in the table? This is just a suggestion, and I leave it to the authors to decide.
4. When citing an author's name, the reference number should be indicated immediately after the name, and not at the end of the sentence.
Author Response
1. It has not been explained on the basis of which three categories of LC scores are defined (median, IQR or?).
Ad 1. The LCs categories cut-off points were defined according to the previous literature (references 4, 7, 9, 28 in our manuscript), based on the total number of B-lines in the lung sonography. We added the paragraph into the Statistical analysis section and the reference in the Results section.
2. In table 2 it would be better if the title of the column called “Statistics” be “p”.
Ad 2. As suggested, we modified the Table 2 titles.
3. Would not it be better to present the results of the correlation described in the text in the table? This is just a suggestion, and I leave it to the authors to decide.
Ad 3. As only several correlations of all analyzed parameters were significant, we decided to present it in a plain text, not in a table.
4. When citing an author's name, the reference number should be indicated immediately after the name, and not at the end of the sentence.
Ad 4. As suggested, the manuscript was modified.
Reviewer 2 Report
This is a study about the value of chest ultrasound as a measure of volume overload in HD patients.
The authors studied 54 HD patients twice (1st and 3rd HD session) and found that lung ultrasound findings are related with overhydration and left ventricle dysfunction.
However both diagnosis should be treated by intensive ultrafiltration!
Another issue is that the patients were used twice but the results are cumulative. What was the rational? They should report each group separately and compare the two groups if there were differences.
The residual dieresis range values do not look correct as there were patients with long HD vintage (101 months!)
Author Response
The authors studied 54 HD patients twice (1st and 3rd HD session) and found that lung ultrasound findings are related with overhydration and left ventricle dysfunction.
However both diagnosis should be treated by intensive ultrafiltration!
Ad 1. This is true that intensive ultrafiltration is usually beneficial both in the case of overhydration and left ventricular dysfunction. However, overhydration results from excessive fluid intake by a patient (in relation to the residual diuresis, if any), whereas LV dysfunction often results from the concomitant heart disease (heart failure, ischemic disease, eccentric hypertrophy, uremic cardiomyopathy, valve defects, amyloidosis), and requires dedicated diagnostics and treatment, including the decreasing of the patient’ “dry” weight.
2. Another issue is that the patients were used twice but the results are cumulative. What was the rational? They should report each group separately and compare the two groups if there were differences.
Ad 2. The study was designed as a non-interventional one. We performed the measurements of fluid status but the results were blinded to the clinician. In a consequence, the ultrafiltration setting was based on daily clinical practice.
The rationale for performing the same ultrasound and BIA measurements in our cohort of hemodialysis patients twice was the fact, that due to the difference in the number of days from the previous dialysis session (3 vs. 2 days) the degree of fluid overload is expected to be higher before first (post-weekend) HD session, as compared to the last session in the week. However, this expected fluid status variability is independent of interpatient differences. This was the reason to treat each assessment independently. In our opinion, the intra-patient comparison could be more valuable in a B-lines-guided interventional study.
3. The residual dieresis range values do not look correct as there were patients with long HD vintage (101 months!)
Ad 3. In Table 1, the residual diuresis is presented as mean values with 95% confidence interval, not as a range. The ranges of residual diuresis values in three subgroups were as follows: group 1: 0-1700 ml, group 2: 0-1500 ml, group 3: 0-1800 ml.
Round 2
Reviewer 2 Report
The explanation about the residual diuresis is at least inappropriate.
There can not be patients with long HD vintage who maintain such a residual renal function. In addition the number provided by the authors in the response are completely different from the text!!!
Author Response
Reviewer 2:
The explanation about the residual diuresis is at least inappropriate.
There can not be patients with long HD vintage who maintain such a residual renal function. In addition the number provided by the authors in the response are completely different from the text!!!
Ad 1.
In our first response, we provided minimal and maximum values (i.e., range) of residual diuresis in the three analyzed subgroups. They were different from values presented in Table 1, because in the Table 1 there were means and 95% confidence intervals (CI).
However, after reevaluation of the manuscript according to the reviewer’s suggestions, we modified the presentation of residual diuresis data, as it was inappropriate due to its skeweness. Instead of means and 95% CI, we now present median values with quartiles (Q1 and Q3). Please note that the median dialysis vintage vary from 21 to 61 months, including also subjects who were hemodialyzed only few months, and their preserved residual diuresis increased both its the mean and the median values. Consequently, we performed the statistics using logarithmized values of residual diuresis.
